# Predictor Selection for Bacterial Vaginosis Diagnosis Using Decision Tree and Relief Algorithms

**Jesús F. Pérez-Gómez** **, Juana Canul-Reich \*** , **José Hernández-Torruco** and **Betania Hernández-Ocaña**

División Académica de Ciencias y Tecnologias de la Información, Universidad Juárez Autónoma de Tabasco, Cunduacán, Tabasco 86690, Mexico; 181H18002@alumno.ujat.mx (J.F.P.-G.); jose.hernandezt@ujat.mx (J.H.-T.); betania.hernandez@ujat.mx (B.H.-O.)

**\*** Correspondence: juana.canul@ujat.mx; Tel.: +52-914-106-3272

**Abstract:** Requiring only a few relevant characteristics from patients when diagnosing bacterial vaginosis is highly useful for physicians as it makes it less time consuming to collect these data. This would result in having a dataset of patients that can be more accurately diagnosed using only a subset of informative or relevant features in contrast to using the entire set of features. As such, this is a feature selection (FS) problem. In this work, decision tree and Relief algorithms were used as feature selectors. Experiments were conducted on a real dataset for bacterial vaginosis with 396 instances and 252 features/attributes. The dataset was obtained from universities located in Baltimore and Atlanta. The FS algorithms utilized feature rankings, from which the top fifteen features formed a new dataset that was used as input for both support vector machine (SVM) and logistic regression (LR) algorithms for classification. For performance evaluation, averages of 30 runs of 10-fold cross-validation were reported, along with balanced accuracy, sensitivity, and specificity as performance measures. A performance comparison of the results was made between using the total number of features against using the top fifteen. These results found similar attributes from our rankings compared to those reported in the literature. This study is part of ongoing research that is investigating a range of feature selection and classification methods.

**Keywords:** feature selection; bacterial vaginosis diagnosis; microbial communities; SVM; logistic regression

## 1. Introduction

Bacterial vaginosis (BV) is a disease affecting millions of women around the world and involves several serious health conditions [1]. It is the most common of the vaginal diseases in women of reproductive age and it is associated with preterm delivery, chorioamnionitis, post-abortion infection, pelvic inflammatory disease, and sexually transmitted diseases, such as human papillomavirus (HPV) [2]. This disease can be detected by two clinical procedures: the Amsel criteria and the Nugent score. Another procedure to detect VB is named real-time or quantitative polymerase chain reaction (qPCR), which consists of the extraction, isolation, and amplification of DNA microorganisms present in the vaginal tract [3]. Some of these procedures usually take a long time to analyze the samples and some are invasive. In diseases like bacterial vaginosis, time is a determining factor for its treatment; therefore, requiring fewer data to diagnose BV is useful for physicians. As such, if there was a more effective and less invasive way to detect this disease, the detection and treatment of BV would be faster and more efficient. According to Liang et al. [4], to formulate a diagnosis, physicians generally ask questions related to the symptoms. From this initial small feature set, the physician forms a differential diagnosis and decides what features (questions, exams, laboratory testing, and/or imaging

studies) to obtain to rule out diagnoses in the differential diagnosis set. Through hypothetic-deductive reasoning, the most useful features are identified such that when the probability of one of the diagnoses reaches a level of acceptability, the process is stopped and the diagnosis is accepted. It may be possible to achieve an acceptable level of certainty of the diagnosis with only a few features and thus not need to process the entire feature set. Therefore, the physician can be considered a classifier of sorts. Machine learning algorithms (MLAs) based on artificial intelligence (AI) methods can also do this. From this approach, the objectives and motivations of this work were established: first, using MLAs to explore the most relevant attributes in the data of BV that minimizes the needed data to diagnose it, and second, to explore the features or feature subsets that positively impact the performance of the classification algorithms (CAs) regarding diagnosing BV. For this reason, this work was addressed as a feature selection problem. Feature selection (FS) involves the reduction of the number of features (attributes, variables, or predictors) for each instance in a dataset, discarding the less important attributes [5]. The main objectives of FS involve improving the prediction performance of the classifiers, providing faster and more cost-effective predictors, and providing a better understanding of the underlying process that generated the data [6]. The support vector machine (SVM) and logistic regression (LR) classifiers were used in this study to measure the ability to classify patients either as positive or negative cases of BV by obtaining the balanced accuracy, sensitivity, and specificity performance measures. The dataset used for this objective was obtained from universities located in Baltimore and Atlanta. It contained clinical and biological information about vaginal microorganisms. Finally, the contribution made by this research can be summarized as follows:

(1)　The determination of the best fifteen predictors for bacterial vaginosis diagnosis using feature selection algorithms.
(2)　Comparison of the results obtained in this research to those obtained in Beck and Foster [7].
(3)　The determination of a highly promising combination of SVM as a classification algorithm and decision trees as a feature selector for bacterial vaginosis diagnosis.

Only a few investigations have been carried out regarding machine learning methods applied in the field of bacterial vaginosis. Some of the works that motivated this research are described below. The purpose of Baker et al.'s [8] work was to discover the most significant features of BV and to apply some classification algorithms to diagnose the BV. Twenty feature selection algorithms, in combination with nine classification algorithms, were applied using Weka. The precision (proportion of correctly classified instances), the recall (proportion of true positive and false negative), and the number of attributes reduced were some of the metrics they considered. They determined that the functional tree (FT) and WrapperSubSetEval algorithms produced the best combination.

Beck and Foster [7] applied random forests (RFs) and LR, in combination with Relief, to diagnose BV. To rank the features, they considered the purity increase in the node as a measure in the RFs. For LR, the features were ranked according to their mean coefficient magnitude in all cross-validation datasets divided by their standard deviation. Relief was implemented to calculate a third feature ranking. A table with the top 15 important features ranked by classification accuracy was obtained. Features like *Aeroccocus*, *Atopobium*, *Dialister*, *Eggerthella*, and *Gardnerella* were categorized as the most important. Their results and the results of this research were compared.

Hall [9] evaluated CFS (correlated-based feature selection) with three machine learning algorithms—C4.5 (decision trees), IB1 (an instanced-based learner), and naïve Bayes—on artificial and natural datasets to test the hypothesis that algorithms based on a correlation between attributes improved the performance of the classifiers. The accuracy, CPU time, and the number of features reduced were used as the performance measures. The experiments on artificial domains showed that the methods were able to identify interactions between features. They concluded that a superior algorithm does not exist for all machine learning problems, but in many cases, the CFS algorithm can enhance the performance of classification algorithms, while at the same time achieve a reduction in the features used.

This work is organized as follows: Section 1 provides the introduction and the current state of the art; Section 2 explains the details regarding the used dataset, methods, and techniques; in Section 3, the results obtained by all phases of the experimental design are provided; and finally, the conclusions are provided in Section 4.

## 2. Materials and Methods

### 2.1. Dataset

The dataset used for this work was generated by Ravel et al. in 2011 [10]. It contains information about vaginal bacterial communities of asymptomatic North American women, of whom 97 were BV+. The sampling was performed at three clinical sites: two in Baltimore at the University of Maryland School of Medicine and one in Atlanta at Emory University. The data extraction consisted of vaginal samples with two self-collected swabs between June 2008 and January 2009 of regularly menstruating, non-pregnant women that were of reproductive age from 12 to 45 years and had a history of sexual activity. Through the swabs, the Nugent criteria and clinical data were obtained. Furthermore, the swabs were submitted to whole-genome DNA extraction to obtain the pyrosequencing of the barcoded 16S rRNA gene amplicon. The product was quantified using BioRad and Quant-iT PicoGreen dsDNA assays. Finally, a dataset with 252 features or columns and 396 instances or rows of microbiological and clinical data of BV were obtained. The dataset is publicly available [10]. The original clinical study is registered at clinicaltrials.gov under ID number NCT00576797.

The dataset was preprocessed as follows. First, all categorical values in the dataset were replaced by integer numbers (e.g., for the Nugent score categories: low = 1, intermediate = 2, and high = 3). Furthermore, explicit class labels of BV+ were given to instances with a Nugent score value greater than or equal to 7; otherwise, the instance classes were set to BV−, according to the author's definition [7]. A summary of the features in the dataset is shown in Table 1.

**Table 1.** Features in the dataset.

| Features | Represents |
|---|---|
| BV | 1 = positive or 2 = negative for bacterial vaginosis. |
| EthnicGroup | Ethnic group to which the test subject belongs. It can take on the values of 1 = Asian, 2 = African American, 3 = Hispanic, and 4 = White. |
| pH | Degree of alkalinity or acidity of a sample. |
| NugentScore | Scoring system for vaginal swabs to diagnose bacterial vaginosis (BV): 7 to 10 is consistent with BV+. |
| NugentScore_Cat | Nugent score grouping according to their values. |
| CommunityGroup | Microbial community to which the test subject belongs. |
| *Megasphaera*, *Eggerthella*, *Lactobacillus crispatus*, and others (247 features) | Count of microorganisms in the vaginal analysis obtained using the qPCR technique on the 16S rRNA gene. |

### 2.2. Feature Selection Algorithms

#### 2.2.1. Decision Trees

Even though a decision tree (DT) is a classifier algorithm, in this work, it was used as a feature selector. This FS algorithm is based on the entropy measure. The entropy is used in the process of the decision tree construction. According to Bramer [5], entropy is an information-theoretic measure of the "uncertainty" contained in a training set. It is calculated using Equation (1):

$$E = -\sum_{i=1}^{K} p_i \log 2 p_i. \tag{1}$$

If there are $K$ classes, we can denote the proportion of instances with classification $i$ as $p_i$ for $i = 1$ to $K$. The value of $p_i$ is the number of occurrences of class $i$ divided by the total number of instances,

which is a number between 0 and 1 inclusive [5]. The lesser the entropy, the greater the information gain of the feature being analyzed [11]. The R package *caret* [12] provides an implementation of the *J48* algorithm for decision trees.

### 2.2.2. Relief

Relief is a multivariate filter-method for feature selection. It selects relevant features based on the differences of feature values between pairs of instances according to the nearest-neighbor algorithm, providing a score for each feature [13]. For a random instance (*Ri*), Relief searches the two nearest neighbors: one from the same class (nearest hit *H*) and the other from a different class (nearest miss *M*). It updates the quality estimation (*W*[*A*]) for all attributes *A* depending on their values for *Ri*, *M*, and *H*. If instances *Ri* and *H* have different values of the attribute *A*, then the attribute *A* separates two instances with the same class such that the quality estimation *W*[*A*] is decreased. On the other hand, if instances *Ri* and *M* have different values of the attribute *A*, then the attribute *A* separates two instances with different class values such that the quality estimation *W*[*A*] is increased. The whole process is repeated *m* times, where *m* is a user-defined parameter [13]. The pseudocode is shown in Algorithm 1.

---

**Algorithm 1**. Relief pseudocode

---

*Input*: a vector of attribute values and the class value for each training instance
*Output*: the vector *W* of estimations of the qualities of attributes
1. set all weights *W[A]: = 0.0*
2. **for** *i: = 1* **to** *m* **do begin**
3. 　　　randomly select an instance *Ri*;
4. 　　　find nearest hit *H* and nearest miss *M;*
5. 　　　**for** *A: = 1* **to** *a* **do**
6. 　　　　　*W[A]: = W[A] − diff(A, Ri, H)/m + diff(A, Ri, M)/m;*
7. **end**;

---

Function *diff* (*A, I1, I2*) calculates the difference between the values of the attribute *A* for two instances *I1* and *I2*. The function *diff* is also used for calculating the distance between instances to find the nearest neighbors and it is based on the Manhattan distance [13]. The R package *FSelector* [14] provides an implementation of the *Relief* algorithm.

### 2.3. Feature Ranking and Cutoff

The feature selection algorithms can include a feature ranking or a variable ranking as an auxiliary selection mechanism [6]. One of its common uses is to discover a set of leading attributes that can later be used to create a subset of the data. A ranking criterion is used to find the most important features that discriminate between classes. Ranking criteria used are the entropy and a distance measure similar to the one used in the nearest neighbor algorithm, which is computed between a pair of instances. A feature subset is defined by applying a cutoff that uniquely considers some features. This cutoff may be determined using a statistical measure or subjective likelihood of relevance, or simply using the desired number of features in the subset [15]. To create our subsets, only the top fifteen features as ranked by both Relief and a DT were considered. This cutoff value was used by Beck and Foster [7], and was used in this work for comparison purposes between their results and those obtained in this research.

### 2.4. Classification Algorithms

Classification algorithms consist of a learning phase, where a classification model is constructed, and a classification phase, where the model is used to predict a label for given data [16]. According to Aggarwal [17], the classification problem may be stated as follows: given a set of training data points,

along with associated training labels, the problem is to determine the class label for an unlabeled test instance. A support vector machine and logistic regression were used as the classification algorithms for this work since authors [7,8] had reported them as good for helping with diagnoses in medical cases. They can be fitted to linearly separable data, which was the case in this research.

### 2.4.1. Support Vector Machine

According to Wang et al. [18], an SVM is an algorithm that creates a model that represents the sample points in space by separating the classes as much as possible in that space. When a new class is evaluated using an SVM based on its proximity to the information in the model, this new class will be classified into one or another class. The SVM algorithm is provided in the *e1071* R software package [19].

### 2.4.2. Logistic Regression

Logistic regression (LR) allows for a categorical response variable to be related to a set of predictor variables, such as the modeling of a numeric response variable, using a linear model [16]. It is used to model the probability of a certain class, label, or existing event. The LR algorithm is provided in *glmnet* in the R software package [20].

### 2.5. K-Folds Cross-Validation

K-folds cross-validation is one of the most frequently used methods for obtaining reliable estimates for small datasets and is also usually called 10-Fold cross validation (10-FCV). Torgo [21] describes it as follows: Obtain k equally sized and random subsets of the training data. For each of these k subsets, build a model using the remaining k−1 sets and evaluate this model on the kth subset. Store the performance of the model and repeat this process for all remaining subsets. In the end, there are k performance measures, all obtained by testing a model on data not used for its construction, which is the key property. According to a review of the state of the art [7,8], the most popular value used for k is 10, which was used in this work.

### 2.6. Performance Measures

### 2.6.1. Balanced Accuracy

The dataset was imbalanced since the number of instances of each class was unequal. Therefore, balanced accuracy was used as a performance measure instead of total accuracy [22] and is referred to as the precision obtained by the classification algorithms. It assesses the accuracy individually for each class without distinguishing between the other classes. Balanced accuracy, also named weighted accuracy, is calculated using Equation (2):

$$\text{Balanced Accuracy} = \frac{\left( \frac{tp}{tp+fn} + \frac{tn}{fp+tn} \right)}{2}, \tag{2}$$

where *tp*, *tn*, *fp*, and *fn* are the true positive, true negative, false positive, and false negative prediction values, respectively, in the confusion matrix. We refer to this performance measure as the accuracy in this paper.

### 2.6.2. Sensitivity

According to Bramer [5], sensitivity is a performance measure that gives the proportion of positive instances that are correctly classified as positive. In medicine, it is interpreted as the confidence level that a test correctly generates a positive result. The sensitivity is calculated using Equation (3):

$$\text{Sensitivity} = \frac{tp}{tp + fn} \ . \tag{3}$$

### 2.6.3. Specificity

Specificity is the proportion of negative instances that are correctly classified as negative [5]. In medicine, it is interpreted as the confidence level that a test correctly generates a negative result. The specificity is calculated using Equation (4):

$$\text{Specificity} = \frac{tn}{tn + fp} \ . \tag{4}$$

### 2.7. Experimental Studies

The experiments were divided into three scenarios, which are detailed below. The first one consisted of 30 runs of each of the SVM and LR classification algorithms using the full features set of the dataset described in Section 2.1. Each run was conducted under a 10-FCV scheme for the performance evaluation. Across the 30 runs, a different seed was used. For comparison purposes between SVM and LR, the same seed was used for the same number of runs. SVM was used with a linear kernel.

The second scenario of experiments consisted of conducting a feature selection process on the dataset before learning any classification model. Relief and decision trees were used as feature selectors, which allowed us to obtain two feature rankings from all the features in the dataset. Subsets with the top 15 features in each feature ranking were created, which were used in the training phase in the classification models. Experiments were similarly conducted to those in the first scenario, that is, within a 10-FCV scheme repeated 30 times with different seeds each time. The feature selector methods were performed on the training phase at each iteration of the cross-validation process.

The third scenario consisted of conducting 30 runs of each classification algorithm using only the fifteen most relevant features obtained in the work of Beck and Foster [7] for comparison purposes.

## 3. Results

Before starting with the experimental phase, the method used to determine the rankings of the top 15 attributes using the FS algorithms is given. Table 2 shows the structure of the results obtained from the runs of the feature selection algorithms. An importance value for each attribute was obtained for each FS method. The mean importance value (MIV), shown as the last column in Table 2, is the base that was used to create our feature rankings. The rows represent the features in the dataset described in Section 2.1. The columns represent the number of runs of the FS methods.

**Table 2.** The mean importance value (MIV) for each feature, which is the average rank of the feature across 30 runs of 10-Fold cross validation (10-FCV). This was performed for each feature selection (FS) algorithm.

| Features | Run1 | Run2 | Run3 | ... | Run300 | MIV |
|----------|------|------|------|-----|--------|-----|
| Var1 | Importance | Importance | Importance | ... | Importance | MIV_Var1 |
| Var2 | Importance | Importance | Importance | ... | Importance | MIV_Var2 |
| Var3 | Importance | Importance | Importance | ... | Importance | MIV_Var3 |
| ... | ... | ... | ... | ... | ... | ... |
| Var252 | Importance | Importance | Importance | ... | Importance | MIV_Var252 |

From this structure and for comparison purposes, Table 3 was created. In this table, the feature rankings obtained using the Relief and decision tree algorithms as feature selectors are shown. The MIV is the average importance value obtained by each feature across all runs of the FS method. The algorithm provided a range of values depending on the technique used to evaluate the features.

**Table 3.** Top 15 relevant features (best predictors of BV) obtained using the FS algorithms. The "Beck and Foster" column was added to compare the results from this work and those obtained by Beck and Foster in 2015 [7]. The features are ordered according to the mean importance value (MIV). Features common to all three rankings are labeled "a". Features common to two rankings are labeled "b".

| Relief | | Beck and Foster | Decision Trees | |
|---|---|---|---|---|
| **Features** | **MIV** | **Features** | **Features** | **MIV** |
| Nugent_score_catb [b] | 0.8113 | *Prevotella* [a] | Nugent_score [b] | 100 |
| Nugent_score [b] | 0.5565 | *Dialister* [a] | Nugent_score_catb [b] | 100 |
| *Prevotella* [a] | 0.2277 | *Gardnerella* [b] | *Prevotella* [a] | 86.12 |
| *Megasphaera* [a] | 0.1899 | pH [a] | *Dialister* [a] | 86.09 |
| CommunityGroupc [b] | 0.145 | *Megasphaera* [a] | *Gardnerella* [b] | 78.53 |
| pH [a] | 0.1414 | *Atopobium* [a] | *Megasphaera* [a] | 75.4 |
| *Sneathia* [a] | 0.1045 | *Eggerthella* [a] | pH [a] | 74.62 |
| *Dialister* [a] | 0.1 | *Sneathia* [a] | *Atopobium* [a] | 74.47 |
| *Eggerthella* [a] | 0.0977 | *Peptoniphilus* [a] | *Eggerthella* [a] | 73.41 |
| *Ruminococcaceae*3 [a] | 0.0925 | *Parvimonas* [b] | *Sneathia* [a] | 72.3 |
| *Lachnospiraceae_8* | 0.0673 | *Ruminococcaceae*3 [a] | CommunityGroupc [b] | 69.79 |
| *Atopobium* [a] | 0.0565 | *Lactobacillus crispatus* | *Parvimonas* | 67.76 |
| *Peptoniphilus* [a] | 0.0501 | *Aerococcus* | *Ruminococcaceae*3 [a] | 66.92 |
| *Bulleidia* | 0.044 | *Ruminococcaceae sedis* | *Peptoniphilus* [a] | 63.85 |
| *Coriobacteriaceae_2* | 0.0401 | *Lactobacillus iners* | *Prevotellaceae_2* | 60.68 |

Both FS algorithms investigated in this work obtained similar results to each other. They shared 12 features in their final rankings. Additionally, the two final rankings obtained had at least eleven features in common with the top 15 obtained by Beck and Foster [7].

As described in Section 2.7, we experimented using three scenarios. In scenario one, 30 runs of the SVM algorithm with a 10-FCV schema using all 252 attributes in the dataset were performed. In scenario two, first, 30 runs of the SVM with 10-FCV were performed using a subset of the top 15 most important features as obtained by Relief; second, 30 runs of the SVM were performed using the subset created with the 15 most relevant features obtained using DTs. In scenario three, 30 runs of the SVM were performed using the top 15 features obtained by Beck and Foster [7]. The mean balanced accuracy obtained in each scenario is shown in Figure 1.

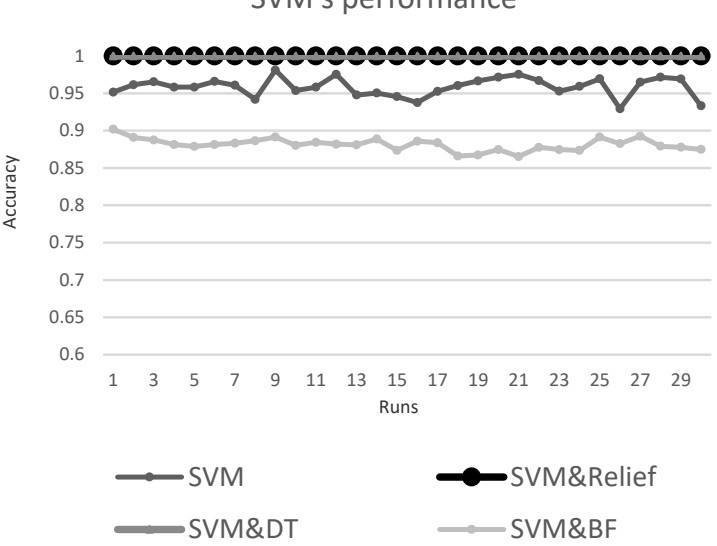

**Figure 1.** Accuracies obtained with the support vector machine (SVM) in all the scenarios. DT: Decision tree, BF: Beck and Foster.

Following previous experiments, we investigated the logistic regression algorithm under the same three scenarios. The mean balanced accuracy obtained in each scenario is shown in Figure 2.

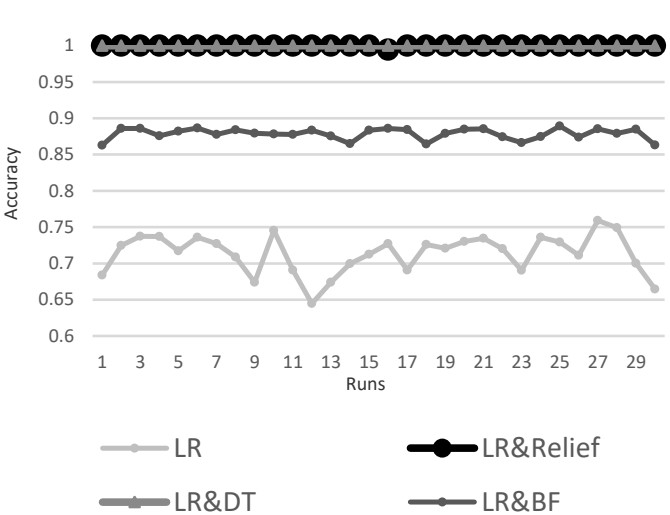

**Figure 2.** Accuracies obtained with logistic regression (LR) in all the scenarios. DT: Decision tree, BF: Beck and Foster.

The mean balanced accuracy, mean sensitivity, and mean specificity performance measures were obtained across all runs by each classifier. These results are shown in Table 4. The rows represent the classifiers used. The columns represent the results of classifiers in each previously described scenario.

**Table 4.** Performance measures obtained using the classifiers in all experiments. Acc is the mean balanced accuracy or precision, Sens is the mean sensitivity, and Spec is the mean specificity.

| Classifier | 252 Features | | | 15 Best Features Using Relief | | | 15 Best Features Using DTs | | | 15 Best Features From Beck and Foster | | |
| --- | --- | --- | --- | --- | --- | --- | --- | --- | --- | --- | --- | --- |
| | Acc | Sens | Spec | Acc | Sens | Spec | Acc | Sens | Spec | Acc | Sens | Spec |
| SVM | 0.958 | 0.988 | 0.928 | 1 | 1 | 1 | 1 | 1 | 1 | 0.881 | 0.957 | 0.805 |
| LR | 0.713 | 0.803 | 0.623 | 0.999 | 0.999 | 1 | 1 | 1 | 1 | 0.878 | 0.955 | 0.801 |

## 4. Conclusions

In this work, the most relevant attributes of a microbiological dataset for the medical diagnosis of bacterial vaginosis were determined using two feature selection methods, namely Relief and decision tree algorithms. After running the experiments, two feature rankings were obtained with the most relevant predictors of BV. The top 15 features from our rankings were compared to the top fifteen obtained by Beck and Foster [7]. It was found that the three rankings had 11 features in common.

The performance of the classification models created using an SVM and logistic regression with the top fifteen features as ranked by a decision tree, Relief, and Beck and Foster's algorithm was calculated, and a comparison was made against the performance of models created using the entire set of features. The results confirmed an improvement in the performance of classification models when created using datasets with a reduced number of features. A combination of a decision tree as a feature selector method and a support vector machine as a classification algorithm produces models displaying promising performance.

Further research is being conducted with other methods of feature selection and other classification algorithms, which include the biological meaning of these attributes for the diagnosis of bacterial vaginosis.

**Author Contributions:** Conceptualization, J.F.P.-G. and J.C.-R.; methodology, J.F.P.-G., J.C.-R., J.H.-T., and B.H.-O.; software, J.F.P.-G., J.H.-T. and J.C.-R.; validation, J.F.P.-G. and J.C.-R.; formal analysis, J.F.P.-G., J.C.-R., J.H.-T., and B.H.-O.; investigation, J.F.P.-G. and J.C.-R.; resources, J.F.P.-G., J.C.-R., J.H.-T., and B.H.-O.; data curation, J.F.P.-G. and J.C.-R.; writing—original draft preparation, J.F.P.-G. and J.C.-R.; writing—original draft, J.F.P.-G.; writing—review and editing, J.F.P.-G., J.C.-R., J.H.-T., and B.H.-O.; visualization, J.F.P.-G.; supervision, J.C.-R.; project administration, J.C.-R. All authors have read and agreed to the published version of the manuscript.

**Funding:** This research received no external funding.

**Conflicts of Interest:** The authors declare no conflict of interest.

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
