# Peer review of "Predictor Selection for Bacterial Vaginosis Diagnosis Using Decision Tree and Relief Algorithms"

_applsci, doi:10.3390/app10093291_

Round 1
Reviewer 1 Report
The paper has an interesting and useful subject connected to the feature selection for medical diagnosis. According to the acknowledgments, the research was supported by Ministry. This means that the authors have been motivated for this work. The question is: Was the study ordered by a medical center or it was made just for the sake of research? The answer could be added to the introduction.
Even if the general aspect of the paper is good, and the concise presentation can help the reader, there are some details that are necessary. Here are several comments, questions and recommendations meant to improve the quality of the paper.
- In the reviewer's opinion, a research paper should have a non-personal aspect. If the authors agree, please try to avoid the expressions that contain “we”, “our” or other personal terminologies. For instance: “we investigated”, “we used”, “we describe”, “our case” and so on.
- The paper contains some mistakes made by negligence. Please, pay attention to them: line 76 “be-tween”; line 77 “no exists”; lines 99 and 100 “pi” should be used instead of “pi”; the second “Figure 1.” is in fact “Figure 2.”.
- The authors used 395 instances. How does this number influence the performances or the results? Are 395 instances enough to draw the conclusions?
- In table 1, the authors describe the features of the database. Two of these features are “ethnic group” and “community group”. It would be interesting to find out what is the meaning of these two features in this context and, more important, what is their influence on Bacterial Vaginosis. There are some ethnic groups or communities that are predisposed to this disease?
- A good and synthetic presentation of materials and methods is made. However, more technical aspects are required in order to connect the presented theoretical elements to the practical domain they have been used.
- In line 184, the authors say “The parameters in R for all classifiers were established by default.”. It is not very clear what is the meaning of that “default” in the current context. Could these parameters have been established in another way?
- More detailed results should be provided.
Author Response
Responses are provided in the attached pdf file.

Reviewer 2 Report
Perez-Gomez et al. have performed research on the predictor selection for bacterial vaginosis using decision trees and relief algorithms. In their work they have used these two machine learning based algorithms and showed that decision tree coupled with support vector machine serving as feature selector and classification algorithm respectively would produce promising outcome in identifying most important features related to the bacterial vaginosis diagnosis. The work has been performed in satisfactory way and used conventional performance matrices to evaluate their results. The presentation is also simple to convey the core message to the readers. This result is recommended to be published.
Author Response
Thanks

Reviewer 3 Report
This manuscript proposed a classification method of bacterial vaginosis by the machine learning of SVM and logistic regression with feature selections such as decision tree, relief, and Beck & Foster's algorithms. It also used the machine learning algorithms of SVM and logistic regression for the Classification. However, many things in the manuscript should be improved and then be resubmitted again after revising wholly in the followings:
- Page 1: Abstract: Please revise the whole abstract to be described research motivation, research goal, research methods and material, experimental results, and conclusion. But the abstract is not complete, so please revise it wholly. Also, replace the keywords to be suitable for your arcticle.
- Page 1-2: 1.Introduction: The introdcution should describe the contents including research motivation, research objectives, and your contribution's summary. Please revise it.
- Pages 2-3: After describing the dataset, data extraction contents should be described clearly, because it is very critical for your experimental results. Sometimes it can be drawn by a figure and its decription.
- Page 5: section 2.6: For the performance measure, precision can be added in your experiement with accuracy, sensitivity, and specificity. Thus, please add the description of precision in section 2.6.
- Page 6: line 196: Please remove it; otherwise move it into the results in Chapter 3.
- Page 6: lines 197-199: Authors told that Wilcoxon test was used to compare some results, but there is no result in Chapter 3. I could not find the results for the Wilcoxon test in the chapter 3 Results. Please solve the issue.
- Page 6: Table 2: Give a key to abrebiation of MIV. Also, Page 7: Table 3: the same as before.
- Page 7: Table 3: The table was given the top 15 most important features for the machine learning algorithms of Relief, Beck & Foster, Decision tree. But it does not match the title of this article as well as description of the feature selection algorithms in section 2.2. Maybe explain the Beck & Foster's algorithm in section 2.2, please.
- Page 8: Tavle 4: Please add the experimental results for precision, as explained before.
- Page 8: Conclusion: Please summary the conclusion in brief within 10-20 lines. The end.
Author Response

(The authors gave the same response as above.)

Round 2
Reviewer 1 Report
The authors have revised their paper according to my comments. I believe the manuscript has been significantly improved and now it can be published in Applied Sciences.
Author Response
Thanks
Reviewer 3 Report
Many things in the manuscript were improved via the revision. It could be accepted after revising the following two things:
- Please replace the Cross Validation in the Keyword with Bacterial Vaginosis Diagnosis.
- Please describe the detailed data extraction and preprocessing methods from the raw data for the experiment clearly.
That is it. The end.
